# Nanovaccines against Viral Infectious Diseases

**DOI:** 10.3390/pharmaceutics14122554

**Published:** 2022-11-22

**Authors:** Wen Tzuen Heng, Jia Sheng Yew, Chit Laa Poh

**Affiliations:** Centre for Virus and Vaccine Research, School of Medical and Life Sciences, Sunway University, Subang Jaya 47500, Malaysia

**Keywords:** adjuvants, nanoparticles, nanotechnology, nanovaccine development, viral infectious diseases, preclinical and clinical trials

## Abstract

Infectious diseases have always been regarded as one of the greatest global threats for the last century. The current ongoing COVID-19 pandemic caused by SARS-CoV-2 is living proof that the world is still threatened by emerging infectious diseases. Morbidity and mortality rates of diseases caused by Coronavirus have inflicted devastating social and economic outcomes. Undoubtedly, vaccination is the most effective method of eradicating infections and infectious diseases that have been eradicated by vaccinations, including Smallpox and Polio. To date, next-generation vaccine candidates with novel platforms are being approved for emergency use, such as the mRNA and viral vectored vaccines against SARS-CoV-2. Nanoparticle based vaccines are the perfect candidates as they demonstrated targeted antigen delivery, improved antigen presentation, and sustained antigen release while providing self-adjuvanting functions to stimulate potent immune responses. In this review, we discussed most of the recent nanovaccines that have found success in immunization and challenge studies in animal models in comparison with their naked vaccine counterparts. Nanovaccines that are currently in clinical trials are also reviewed.

## 1. Introduction

Infectious diseases have posed one of the greatest threats to public health globally for the past few decades. The most recent influenza A(H1N1) pandemic2009 caused 60.8 million infections and 12,469 deaths in the United States [1]. The acquired immunodeficiency syndrome (AIDS) pandemic has dramatically changed the global burden of infectious diseases since the early 1980s, yet an estimated 38 million people still live with HIV at the end of 2019 [2]. To date, the emergence of the new coronavirus disease-2019 (COVID-19) pandemic has rapidly spread over the world, infecting 615 million people and causing over 6.5 million mortalities as of October 2022 [3]. Control and prevention of these infectious pathogens by vaccinations are undoubtedly the most appropriate way of medical interventions for the control of infectious diseases. Conventional vaccines have been developed against infectious diseases, and these vaccines usually consist of inactivated or live-attenuated microorganisms for the induction of immune responses that could confer protection to the hosts. For instance, one of the deadliest diseases known to humans—smallpox, is the only human disease that had been eradicated through the prophylactic administration of the live vaccinia virus (VACV) [4]. Conventional vaccines are usually composed of whole pathogens or several pathogens that might carry excessive antigenic loads, which increases the risk of reactogenic effects. As such, autoimmune and severe allergic responses such as anaphylactic shock are the main concerns associated with attenuated or inactivated vaccines [5]. Egg allergy has been of concern for influenza vaccines as egg-based vaccines might contain small amounts of residual egg proteins (ovalbumin) during the manufacturing process [6]. Moreover, neurological disorders in children associated with live attenuated MMR vaccine—a vaccine against measles, mumps, and rubella viruses were reported in 2002. Although the establishment of severe measles inclusion-body encephalitis (MIBE) after MMR vaccination is rare, the mortalities associated with such disorders were reported at 10–20% [7]. Despite these limitations, there are several LAIV vaccines that have been licensed against infectious agents, including influenza, measles, rotavirus, yellow fever as well as the oral polio [8,9,10]; and inactivated vaccines against rabies and influenza [11,12].

Since the onset of the COVID-19 pandemic, the development of next-generation vaccine candidates has been moving forward at warp speed. This has highlighted the necessity to use novel vaccine platforms to control and prevent COVID-19. These vaccine candidates would allow for more clinical evaluations and regulatory choices in phase IV clinical trials to accommodate the differences in immunological responses. To date, several novel vaccines, including mRNA-based vaccines (Pfizer, Moderna) and the viral vectored [AstraZeneka, Janssen (Johnson & Johnson)] vaccines, were granted for human use under emergency approval [13]. The advantages of these novel vaccines are that they can be produced without handling the live virus during production, and they can induce robust immunity in the vaccinated individuals [14]. The mRNA-based vaccine gained the most attention due to its safety profile that offers minimal risk of infection and insertion mutagenesis. mRNA vaccine could be manufactured in a rapid and scalable manner owing to their in vitro cell-free transcription reactions, which minimized the potential of a cell or viral contaminations [15]. In addition, modifying the structural features of the mRNA-based vaccine could enhance the stability and immunogenicity of the vaccine candidate. For instance, Wayment-Steele et al. (2021) demonstrated that modification of the secondary structure of the mRNA improved its stability against endonuclease cleavage and chemical degradation [16]. Similarly, a two-dose intramuscular immunization of modified mRNA encoding Zika Virus (ZIKV) prM-E successfully induced high levels of neutralizing antibodies that protected the immunized mice with acquired innate immune deficiencies against ZIKV infection [17]. Due to its ability to induce both humoral and cellular mediated immunity, mRNA-based vaccines are now the most accepted vaccine platform in the development of vaccines against viral infectious diseases [18,19,20]. It is noteworthy that mRNA is highly susceptible to nuclease degradation; hence it usually requires an effective delivery system that ensures its targeted delivery to induce robust immune responses [21]. A nucleoside-modified mRNA encoding the broadly neutralizing anti-HIV antibody, VRC01, was designed and delivered by using lipid nanoparticles (LNP). A single-dose immunization with the vaccine candidate protected the humanized mice from intravenous HIV- infection, demonstrating the nucleuoside-modified mRNA-LNP platform is a safe and effective alternative to conventional vaccines [20].

Apart from that, other vaccine platforms such as recombinant protein, DNA, and peptide-based vaccines have become more attractive due to their safety and feasibility for mass production. DNA-based vaccines usually contain genes of interest that encode the immunogenic antigens and have to be translocated into the nucleus of the host cell by using a plasmid as a vector. This approach aims to induce humoral and cellular-mediated immune responses efficiently. Although they have yet to be approved for human use, several clinical trials of DNA vaccines in clinical phases II and III against COVID-19 were reported to have promising results regarding their safety and efficacies [22]. Notably, the ZyCoV-D DNA vaccine developed by Zydus Cadila from the Department of Biotechnology, Government of India, is the first DNA-based COVID-19 vaccine granted by The emergency use authorization in India. This three-dose DNA vaccine consisting of p-VAX-1, a standard commercially available plasmid vector encoding the Wuhan Hu-1 spike antigen of SARS-CoV-2 S and an IgE signal peptide which is administered intradermally by the needle-free PharmaJet Tropis device. In the clinical phase III trial, the vaccine candidate was found to be safe with an approximately equivalent prevalence of solicited adverse events between the ZyCoV-D and the placebo groups. Apart from that, it was reported that the vaccine yielded a vaccine efficacy of 66.6% in the vaccinated individuals with aged >12 years during the outbreak caused by the Delta variant in 2021 [23]. Other than that, there were several DNA vaccines for COVID-19 that are being evaluated in clinical trials phase 2/3 with the AG0302-COVID-19 developed by AnGes, Osaka University (NCT04527081); INO-4800 developed by Inovio Pharmaceuticals, Advaccine Biopharmaceuticals (NCT04642638); and GX-19N by Genexine (NCT 05067946) being the frontliners [24].

On the other hand, the peptide-based vaccine contains epitopes corresponding to only a fraction of the entire antigen, which capable of inducing desirable B and T-cell immune responses [25]. These peptides comprise 15–30 amino acids that contain epitopes as the immunogenic determinants for the activation of the appropriate cellular and humoral immunity [26,27]. They usually contain potential B and T-cell epitopes with eight to ten amino acids for CD8^+^ T cell activation, twelve to sixteen amino acids for activation of CD4^+^ T cells, and an average of 15 amino acids for the activation of B cells [28,29]. The peptide-based vaccine is an alternative approach to overcome issues of possible side effects related to living or multicomponent vaccines as they could be manufactured with no risk of reactogenic or allergenic effects [30]. In addition, they could be designed to carry multi-epitopes (B and T-cell epitopes) to elicit specific humoral and cellular-mediated responses.

Although DNA and peptide-based vaccines are cost-effective and have minimal risk of infection as well as the ability to elicit specific immunity against pathogens, there are a number of limitations associated with the effective delivery of these vaccines to the targeted sites. Thus, the use of an adjuvant and delivery system consisting of biomaterials is often necessary to potentiate the immunogenicity to provoke potent immune responses.

## 2. Nanotechnology-Based Delivery System

Traditionally, vaccines were delivered through the parenteral route, including the intramuscular (IM) and subcutaneous (SC) pathways (Figure 1). The SC immunization route provided better vaccine drainage to the lymph nodes and stronger immunogenicity compared to the IM pathway. However, the IM administration route was reported to have less severe local side effects than the SC immunization route despite the presence of an adjuvant in the vaccine formulation [31]. Intravenous delivery of nanoparticles is advantageous due to its ability to deliver efficiently through wide range distribution in the systemic circulation and rapid response. IV administration also bypasses the risk of degradation by proteolytic enzymes and first-pass metabolism [32]. The first pass phenomenon is when the delivered drug/vaccine reaches a major site with metabolically active tissues such as the GI tract or liver and results in a reduced concentration of target vaccine/drug upon distribution to the site of action [33].

Administrations of vaccines through the oral or nasal route are also attractive alternatives as both the GI tract and nasal mucosa are highly vascularized, thus allowing the induction of both systemic and mucosal immunity [34,35]. Additionally, it is desirable for the immunization route to target respiratory diseases to generate robust mucosal and systemic immunity. For example, oral vaccine administration targets hand food and mouth disease (HFMD) transmitted through the fecal-oral route and intranasal vaccination targets respiratory diseases such as influenza A [35,36,37]. Nasal delivery also has the advantage of minimal systemic side effects due to the large surface area of the lungs, leading to nanoparticles accumulating within pulmonary tissues and avoidance of first-pass metabolism [38,39]. The live attenuated influenza vaccine (LAIV) delivered through spraying is considered to be advantageous due to its low invasive administration route [40]. Another non-invasive method for nanoparticles to gain access to the systemic circulation is by transdermal penetration of the human skin after topical application, which also bypasses first-pass metabolism [41].

Recently, the advancement of nanotechnology has facilitated the development of next-generation vaccines in which nanoparticles are utilized for delivering and improving the efficacy of vaccines. These nanoparticles can be defined as nanoscaled-sized particulates that mimic the structural features of the viruses, which make them ideal candidates for the development of next-generation vaccines [42]. There are several strategies in which the vaccines can be delivered by nanocarriers, such as encapsulation of the antigens within the nanocarriers or conjugation on the surface of the nanocarriers. Encapsulation of the antigens within the nanocarriers could protect the encapsulated antigens from enzymatic degradation, and conjugation of the antigens on the surface of the nanocarriers could enhance antigen uptake by targeting specific immune cells. In addition to their delivery functions, the nanocarriers displayed intrinsic adjuvant effects by acting as activators of immune cells for the development of more effective vaccines [43]. They could be designed to enhance vaccine efficacy via controlled or sustained release of antigens over a prolonged period and improve the stability of the antigen by preventing it from degradation. In addition, decorating the surface of the nanovaccine with specific ligands such as Fc receptor, C-type lectin receptor, and anti-DEC-205 (mAb) have been explored as they bind actively to the DC receptors [44,45,46]. Incorporating immunostimulatory molecules in nanoparticles could improve the targeted delivery of antigens and enhance the immunogenicity of the nanovaccine by inducing a specific immune response [47]. Some of the most popular immunostimulatory molecules, including mannose, β-Glucans, chitosan, glycolipids, and saponin, have been studied extensively to enhance he proliferation of antigen presentation of innate immune cells [48]. The α-galactosylceramide (α-GalCer) has recently attracted attention as an immunostimulatory molecule that could be incorporated in vaccine formulation to generate robust immune response [49]. α-GalCer is a synthesized glycolipids that can be recognized by the CD1 receptor on APCs to facilitate the activation of invariant natural killer T-cells (iNKT) [50]. Lipid antigens presented by the CD1 receptor on APCs facilitated the proliferation of iNKT cells and triggered the production of Th1 cytokines such as tumor necrosis factor (TNF), interferon-gamma (IFN-γ), and Th2 cytokines (IL-4, IL-10, IL-13) that in turn induced the proliferation of the cell-mediated and humoral immune responses [51,52,53,54]. It has been shown that iNKT cells possess antiviral potential against viral infections. For instance, a previous study demonstrated tissue-protective cytokine IL-22 secreted by iNKT cells reduced pulmonary epithelial damage during IAV infection [55]. Another study highlighted that Th1 cytokines, including IL-2, TNF-α and IFN-γ secreted by iNKT cells were associated with lower plasma viral loads in non-progressive HIV-I infected individuals, suggesting the effector role of iNKT cells in modulating both innate and adaptive immune responses [56]. Owing to their immunomodulating properties, specific ligands targeting iNKT cells have been exploited for the development of nanovaccines against viral diseases. Khan et al. (2022) showed that liposome co-encapsulating MERS antigen in α-GalCer-bearing liposomes successfully elicited higher production of IgG and IgG2a in the immunized mice when compared to those mice immunized with antigens emulsified in alum adjuvant or liposome encapsulated MERS antigens. In addition, a robust cellular mediated immune response was induced by the immunization with the nanovaccine, as indicated by the increased levels of IL-4 and IFN-γ in murine splenocytes [57]. It highlighted that co-encapsulation of iNKT-specific ligand (α-GalCer) in nanoparticles could enhance the immunogenicity of the nanovaccine by eliciting antigen-specific CD4^+^ and CD8^+^ T-cells through the secretion of Th1 and Th2 cytokines.

In order to design a safe and effective nanovacine, it is imperative to understand the mechanisms mediated by the nanovaccine in both innate and adaptive immune responses.

## 3. Activation of Innate and Adaptive Immunity by Nanovaccines

The innate immune system serves as the first line of defense by generating a non-specific inflammatory response upon the detection of invading pathogens, often associated with bacteria or viruses. The complement system is an important part of the innate immune system, and it is composed of more than 30 plasma proteins that interact with each other in a cascade manner. The innate immune system is made up of different tissue-specific cell types such as natural killer cells (NK), granulocytes, and antigen-presenting cells (macrophages and dendritic cells). These innate immune cells, particularly antigen-presenting cells, express pattern-recognition receptors that can recognize pathogen-associated molecular patterns (PAMPs), resulting in antigen uptake and the generation of inflammatory responses by the secretion of cytokines [58]. These PRRs are expressed either on the cell membrane, such as Toll-like receptors (TLRs) and C-type lectin or (CLRs), or NOD-like receptors (NLRs) and RIG-like receptors (RLRs), which can be found in the cytoplasm. These PRRs are known to promote phagocytosis through binding to a specific molecular pattern, such as bacterial lipopolysaccharide and peptidoglycan molecules, triggering the activation of downstream signaling pathways by producing different types of cytokines and chemokines and resulting in the initiation of the innate immune system and subsequently activate the adaptive immune response [59].

Upon nanovaccine administration, the nanovaccine is recognized as a foreign substance by the innate immune system, leading to the generation of specific immune responses based on their physiochemical properties. The physiochemical properties of nanovaccines, for instance, surface charge, size, and shape, would affect their interactions with the soluble proteins, APCs, and innate immune cells, in particular, affecting cellular uptake and inflammatory responses, followed by other immunoregulatory events. The size of nanoparticles is a key factor in determining the efficiency and mode of cellular uptake. Nanoparticles (NPs) technically range in size from 1 to 1000 nm, and NPs with a diameter, ranging from 20 nm to 200 nm could drain into the lymph nodes, which would be readily endocytosed by the resident dendritic cells, whereas NPs with larger particle size, ranging from 500 nm to 1000 nm, would be taken up by the migratory dendritic cells [60,61]. The size of nanoparticles was also found to play a significant role in the determination of the uptake pathways. Nanoparticles with an average size of 200nm are usually internalized through clathrin or caveolin-mediated endocytosis, whereas nanoparticles with an average size larger than 500 µm would be taken up by micropinocytotsis or phagocytosis (Figure 2) [62,63,64]. It was reported that large-sized nanoparticles tend to favor humoral immune responses, whereas smaller-sized nanoparticles would promote cellular-mediated immune responses. A previous study has demonstrated that PLGA encapsulating BSA with different-sized NPs, ranging from 200 nm, 500 nm, and 1000 nm, stimulated different immune responses. PLGA encapsulating BSA with a size of 1000 nm elicited a greater IgG response than that of the 200 nm and 500 nm NPs [65]. This was attributed to the smaller nanoparticles being efficiently taken up by the APCs for antigen processing and presentation, whereas larger nanoparticles would not be taken up by the APCs but adhered to the surface of the APCs for B-cell activation [65,66]. Although several studies have reported that smaller-size NPs could be considered to be more effectively taken up by the APCs due to their ability to permeate biological barriers, the correlation between the size and the immunological responses has not been evaluated. Hence, it is hard to conclude the optimum size ranges of NPs that could generate the strongest immunological responses.

The shape of nanoparticles also has a significant effect on the interaction with innate immune cells. Different shapes (stars, rods, triangles) of methylpolyethylene glycol coated-anisotropic gold nanoparticles demonstrated different rates of antigen uptake by murine macrophages—RAW263.7 cells. It was found that the triangle shape of gold nanoparticles was the shape with the most efficient cellular uptake, followed by gold nanorods and gold nanostars. The higher cellular uptake of the gold nanotriangle could be due to their larger surface-to-volume ratio, which increased the contact area and the cell membrane, whereas the multiple branches and length of the gold nanostar have to overcome a higher membrane barrier, resulting in lower cellular uptake [67]. Similarly, another study reported that PLGA nanorods showed the highest cellular uptake by both Caco-2 cells and Caco-2/HT-29 co-cultures than the PLGA nanosphere and PLGA nanodisc. The authors explained that this could be attributed to the larger contact surface area in rods for interaction and adhesion with the cell membrane, thereby presenting more sites for cellular uptake [68]. Apart from that, it was reported that the shape of nanoparticles determined the localization of nanoparticles inside the cells. Xu and others (2008) investigated the impact of the shape of nanoparticles on cellular uptake by synthesizing layered double hydroxide (LDH) nanoparticles with different morphology- rods and hexagonal sheets. Although both nanorods and nanosheets were internalized via clathrin-mediated endocytosis, nanosheets were retained in the cytoplasm of the cells, whilst nanorods were found to be localized in the nucleus [69]. This study highlighted the importance of improving antigen processing and presentation to immune cells, which can be achieved by modifying the shape of the nanoparticles.

Another critical parameter affecting the uptake of nanoparticles by the APCs is the surface charge of NPs, which first affects the interaction between the cellular membrane receptors and the NPs, followed by their uptake by the APCs. Cationic NPs were internalized by APCs more rapidly due to the electrostatic attractions between the positively charged NPs and the negatively charged cell membrane, thereby facilitating the endocytosis process [70]. Cationic nanoparticles were shown to be preferentially taken up by the two important lung dendritic cells, CD103 and CD11b, and promoted cellular mediated responses indicated by an increase in the production of chemokines CCL2 and CXCL10 in comparison to the anionic NPs in the lungs [71]. Similarly, the immune responses induced by the PLGA-encapsulated ovalbumin with different surface charges were investigated, and the results demonstrated that both PEI-coated nanoparticles promoted cytoplasmic antigen delivery and triggered the activation of DCs in the lymph node in comparison to the negative-charged NPs [72]. However, not all cationic NPs demonstrated higher antigen uptake; indeed, the surface charge of NPs was found to be specific to certain cell types. This is supported by a previous study that illustrated that the mesoporous silica nanoparticles (MSNs) were efficiently taken up by human mesenchymal stem cells (hMSCs), but an inhibitory uptake effect was observed in 3T3-L1 cells [73]. Altogether, these studies highlighted the importance of the surface properties of NPs in modulating immunological responses.

After internalization, DCs concurrently undergo activation and migrate to the secondary lymphoid organs, such as spleens and lymph nodes, where they present the antigenic peptides to T-cells to initiate antigen-specific immune responses. As such, the activated DCs expressed the processed antigenic peptides on the surfaces bound to the MHC I or II molecules for recognition by antigen specific CD4^+^ and CD8^+^ T cells. Depending on the types of cytokine secretion, Th populations were divided into T-helper type-1 (Th1) and T-helper type-2 (Th2) cells. After the nanovaccine was captured by the APCs and processed in the endo/lysosomal compartments, the antigens released from the nanomaterials were processed into peptide fragments and directed to the surface of the cell membrane to be loaded onto MHC II molecules [74]. The resulting peptide-MHC class II complexes were recognized by the helper CD4^+^ T-cells, and the secretion of cytokines, including IL-4, IL-5, IL6, IL-9, and IL-10 would trigger the proliferation of B-cells during the humoral immune responses [75]. The activated B-cell population would then proliferate in the geminal center and undergo somatic hyper mutations to become specific antibodies against target antigens or remain dormant as memory B cells for a future encounter with the homologous antigens [76]. For antigen cross-presentation in DCs, nanomaterials carrying antigens would be processed and degraded by the proteasome. The antigens that were generated by the proteasome are then transported by the transporter associated with antigen processing (TAP) to the endoplasmic reticulum (ER), and they would be loaded on MHC class I molecules, resulting in an MHC-I peptide complex [77]. The peptide-MHC class I complexes interacted directly with the naïve CD8 T-cells, and the secretion of cytokines such as IL-1, IL-2, IFN-γ and TNF-α drove the Th1 response which in turn stimulated the proliferation of cytotoxic T lymphocytes (CTLs) and strengthened the function of cellular mediated immune responses [78]. On the other hand, antigens internalized by endocytosis are degraded by the lysosomes, and they are processed and become bound to the MHC class II molecules to be presented to the T-cells (Figure 2).

The goal to induce robust T-cell immunity against viral infections through vaccination is dependent on the induction of a specific cytotoxic T-lymphocyte (CTL) response. Upon vaccination, the internalized antigens can be presented via two pathways which include the ‘vacuolar’ pathway and the ‘cytosolic’ pathway. In the ‘vacuolar’ pathway, peptide antigens are processed and degraded in endosome/lysosome compartments to be loaded onto MHC I molecules for antigen presentation. In the ‘cytosolic’ pathway, the antigens were processed by the proteasome in the cytosol for cross-presentation to CD8^+^ T-cells [79]. Despite several studies which have indicated that some antigens could be cross-presented by the ‘vacuolar’ pathway, most studies reported that the translocation of exogenous antigens from the lysosome to cytosol was inefficient, resulting in ineffective cross-presentation [80]. Hence, nanoparticles could serve as the perfect vaccine carrier as they can be easily modified to enhance the cross-presentation of antigens on MHC class I molecules via endosome escape and lysosomal processing mechanism. A recent study by Jiang et al. (2018) modified the aluminum hydroxide adjuvant from gel form into AlO(OH)-polymer nanoparticles which successfully enhanced the ‘cytosolic’ delivery and cross-presentation of the encapsulated antigens. Robust CD8^+^ T cell immunity was induced in comparison to the antigens formulated with CpG adjuvant (Figure 3) [81].

## 4. Lipid-Based Nanoparticles

Lipid nanoparticles are spherical vesicles that are composed of ionizable lipids, phospholipids, cholesterol, and polyethylene glycol (PEG), and they are commonly used for the encapsulation of DNA/mRNA. Upon internalization, these nanoparticles became positively charged at low pH and promoted the interaction with the negatively charged lysosomal membrane, thereby facilitating the release of the encapsulated mRNA into the cytoplasm for protein translation. At physiological pH, these nanoparticles became neutrally charged, and the toxicity effects were reduced [82]. Lipid-nanoparticles have been studied extensively for the past decades for the delivery of drugs due to the ease of production without the requirement of using live cells. The latest successful use of lipid NPs as the vaccine delivery system in the two mRNA-based vaccines against SARS-CoV-2, such as the Moderna (mRNA-1273) and Pfizer (BNT162b2) vaccines. These vaccines were granted emergency approval by the FDA against SARS-CoV-2 for individuals above 18 years of age or older in 2020 [83]. The composition of the lipid nanoparticles used in the two vaccines was very similar as they both contained ionizable lipids which were positively charged at low pH (Figure 4) [84,85]. For mRNA-1273, the lipid NP was used to encapsulate the nucleoside-modified mRNA that encoded the SARS-CoV-2 spike (S) glycoprotein. The results from clinical trial phase III revealed that the vaccine candidate was able to confer 94.1% protection against the Wuhan-Hu-1 SARS-CoV-2 in vaccinated individuals without showing any local and systemic toxicity [86]. On the other hand, BNT162b2 is an mRNA-based vaccine composed of nucleoside-modified mRNA encoding a P2 mutant spike protein encapsulated in lipid nanoparticles. In the clinical phase III trial, BNT162b2 was able to provide 95% protection against COVID-19 in individuals over 16 years of age [87]. With the recent emergence of the Delta variant that was the dominant SARS-CoV-2 VOC circulating in many countries, both of the vaccine candidates, mRNA-1273 and BNT162b2, were effective in preventing Delta hospitalizations and reduced mortality rate to 93.4% and 96.1%, respectively, despite their low efficacies in preventing Delta infections (<50%) [88].

Recently, the modified mRNA-1273 vaccine was designed and used as a booster vaccine with the aim of enhancing the effectiveness of the vaccine candidate against the SARS-CoV-2 variants. mRNA-1273.351 is a modified mRNA vaccine that encoded the S protein derived from the B.1.351 variant and mRNA-1273.211. It is a multivalent mRNA vaccine comprising a mixture of 1:1 mRNA-1273 and mRNA-1273.351. Participants in the clinical study received the booster vaccinations approximately 6–8 months after their primary vaccinations, and the results from the phase II clinical trials revealed that the pseudovirus neutralization (PsVN) titers of the vaccinated individuals were boosted against all variants, including the wild-type original strain, B.1.351, and P.1 variants and the levels of antibody elicited by mRNA-1273 and mRNA-1273.351 were similar or higher than the antibody titers after the primary series of vaccinations. It is noteworthy that the mRNA-1273.351 demonstrated higher neutralization titers against B.1.351 variant when compared to the mRNA-1273 as a booster vaccine [89]. To date, the safety and effectiveness of mRNA-1273.211 are being evaluated in a clinical phase II trial with the clinical identifier (NCT04927065).

In addition to the BNT162b2 and mRNA-1273 vaccines, other LNP-based vaccine candidates that were progressing in clinical trials are presented in Table 1. A lipid NP encapsulating mRNA vaccine, denoted as ARCT-021, was demonstrated to be safe and well-tolerated at multiple dose levels, ranging from 1–10 µg in clinical phase I/II trial (clinical identifier: NCT04480957). In early 2020, McKay and others investigated the immunogenicity of lipid nanoparticles encapsulating a self-amplifying RNA encoding the SARS-CoV-2 spike protein nanovaccine in the murine model. Immunizations of mice with the nanovaccine induced specific SARS-CoV-2 antibodies as well as neutralization antibodies to both the pseudo-virus and wild-type virus. In addition, a potent cellular-mediated response was observed, which was characterized by high levels of IFN-γ production upon re-stimulation with the SARS-CoV-2 peptides [90]. The efficacy of this nanovaccine candidate was investigated in clinical trial phase I with the clinical identifier (ISRCTN17072692). Other vaccine candidates, such as ChulaCov19 developed by the Chulalongkorn University, Thailand, and the CoV-2 mRNA vaccine developed by Shulan, Hangzhou Hospital, have also progressed to phase I clinical trials. The safety profile, tolerance, immunogenicity, and dosage of the nanovaccine were evaluated in adults over 18 years of age (Table 2).

## 5. Liposomes

Lipid-nanoparticles can be divided into five categories depending on their physiochemical properties (i) liposome- composed of a phospholipid, cholesterol, and essential oils; (ii) niosomes—consisting of non-ionic surfactants and cholesterol; (iii) transfersomes—formed by phospholipids, cholesterol and edge activators; (iv) solid lipid nanoparticles (SLNs)—solid spherical nanoparticles made of solid surfactant with a solid lipid core; and (v) nanostructured lipid carrier (NLCs)—composed a liquid lipid core surrounded by surfactant outer layer [91]. Amongst the LNPs, liposomes have been studied extensively for over 30 years and have been approved by the FDA for drug delivery in the 1990s. For instance, Doxil^®^ was used for the clinical treatment of ovarian and metastatic breast cancers [92]. The cationic liposomes were composed of amphiphilic phospholipids and cholesterol, which self-assemble into bilayers with an aqueous core. Hydrophilic antigens such as peptides, nucleic acids, and proteins could be encapsulated in the aqueous core, whereas the hydrophobic antigens such as emulsions, lipopeptides were entrapped within the phospholipid bilayer [93]. Early studies showed the immunomodulatory effects of liposomes, including depot formation and enhanced uptake of antigens by APCs, thereby inducing robust immunity.

Carroll et al. (2018) demonstrated vaccination with H1N1, or H3N2 inactivated influenza virus adjuvanted with cationic liposome-DNA complexes (CLDC) conferred protection in rhesus macaques against IAV H1N1 viral challenge. The virus titers in the trachea of the vaccinated animals were significantly lower compared to un-adjuvanted animals. Heterosubtypic protection by H3N1 inactivated vaccine adjuvanted with CLDC were associated with significantly higher levels of antibodies in the CLDC vaccinated groups [94]. A liposome-based nanovaccine consisting of highly conserved B- and T-cell epitopes derived from IAV was administered intranasally in pigs with the monosodium urate (MSU) adjuvant and was demonstrated to provide partial protection of the vaccinated pigs against the highly infectious H1N1 SwIAV. Reduced lung pneumonia lesions, improved T-cell cytokine secretions, and increased mucosal IgA antibodies in the respiratory tract, which resulted in 8–16-fold reductions of nasal virus shedding when compared to the mock group, were reported [95]. Apart from influenza, liposome has been studied as a delivery vehicle against SARS-CoV-2. In a preclinical study, a biomimetic nanovaccine utilized liposome as a vaccine carrier to encapsulate poly (I:C) which mimicked the viral double-stranded RNA. The nanovaccine was coated with the receptor binding domains (RBDs) of SARS-CoV-2 to mimic the viral structure of the SARS-CoV-2 virus. The immune responses of vaccinated mice were evaluated, followed by pseudovirus challenge, and it showed that mice vaccinated with the bionic-virus nanovaccine were protected from the viral infections, and a significant mucosal immunity was elicited, indicated by the increased levels of IFN-γ and TNF-α in the BALF and the higher titers of sIgA in the respiratory tract of the immunized mice. In addition, nanovaccine delivered intranasally enhanced antigen uptake by dendritic cells and increased immune cell recruitment to draining lymph than the mice administered vaccine via intramuscular and intraperitoneal injections [96].

## 6. Biodegradable Polymeric-Based Nanoparticles

Biodegradable polymeric NPs have been widely investigated in medical applications due to their excellent biodegradable and biocompatible characteristics. These polymeric nanoparticles are able to provide a sustainable release of antigens and improve the stability of antigens from in vivo enzymatic degradation. They can be designed to encapsulate the antigen of interest, or the antigen could be surface adsorbed on the nanoparticles to enhance antigen uptake and presentation to the immune cells. There are two main types of polymeric nanoparticles used in the development of delivery carriers for vaccines, such as chitosan and poly (lactic co-glycolide acid) (PLGA), which have shown great potential in several pre-clinical studies.

## 7. PLGA

Polylactic-*co*-glycolic acid (PLGA) is one of the most widely explored polymers in vaccine and therapeutics delivery due to its high biocompatibility, biodegradability, and easily tunable mechanical properties. It is approved by the US Food and Drug Administration (FDA) and European Medicine Agency (EMA) for medical applications. The payload of PLGA is released slowly through hydrolysis after an initial burst release. Upon administration, PLGA particles will undergo degradation by bulk erosion in which the water will diffuse into the polymeric matrix and hydrolyses the ester bond throughout the polymeric matrix. During this process, the PLGA particles will be degraded to their original monomers, such as lactic acid and glycolic acid, which are by-products of the glycolytic pathway, forming non-toxic, biologically compatible, and metabolizable moieties which can be eliminated from the body [97]. Several studies have indicated that long-lasting immunity can be achieved by the administration of PLGA particles as the vaccine carrier.

For instance, a single immunization in Aotus monkeys with the SPf66 malaria subunit vaccine encapsulated in the PLGA microspheres was shown to confer protection in immunized monkeys by eliciting remarkably high neutralizing antibody levels [98]. Zhao et al. (2013) conducted a study with Newcastle disease virus (NDV) and showed that the PLGA encapsulated DNA vaccine (pFNDV-PLGA-NPs) was able to provide 100% protection to vaccinated chickens in the lethal challenge [99]. Plasmid expressing the F gene of NDV was encapsulated in PLGA nanoparticles and administered intranasally to chickens in one group, while the other group was administered with the naked plasmid DNA by intramuscular injection. Despite the IgG levels from the pFNDV-PLGA-NP group peaked at one week later than the group vaccinated with the naked plasmid, the pFNDV-PLGA-NP group was able to maintain peak levels of IgG for a much longer duration than the chickens that received only the naked plasmid. This indicated that a stronger humoral immunity was induced through intranasal administration of pFNDV-PLGA-NPs, and controlled release effects were observed. In-vitro release analysis of the pFNDV-PLGA-NPs also supported the claim as burst release was observed in the first 48 h, followed by the slow release of high-level DNA payload over 10 days. Lymphocyte proliferation assays indicated that the immunized chickens showed no significant differences in the second week post-immunization. The stimulation index of chickens receiving pFNDV-PLGA-NPs was, however, significantly higher than chickens receiving the naked plasmid in the fourth week post-immunization. These findings indicated that the pFNDV-PLGA-NPs were able to induce robust humoral and cellular-mediated immunity.

A protective humoral and cellular immunity was achieved by a single dose of HBsAg adsorbed on PLGA microspheres (MPs) (Figure 5ii) in comparison to those delivered via three injections of the aluminum adjuvanted HBsAg vaccine in guinea pigs [100]. Hiremath and colleagues (2016) demonstrated that pigs vaccinated with PLGA NPs encapsulating inactivated H1N1 influenza M2e peptide antigens and adjuvant-containing *Mycobacterium vaccae* whole cell lysates showed no fever and flu symptoms after challenge with the swine influenza viruses (SwIV). The peptides encapsulated in PLGA NPs induced a significant increase in CD4^+^ and CD8^+^ T cells that resulted in a reduction in the viral lung loads and reduced the flu symptoms of the vaccinated pigs. Surprisingly, the PLGA NP formulations with and without adjuvant did not induce significant antibody responses, and this could be attributed to the poor adjuvant effect of *Mycobacterium vaccae* or the uneven distribution of the PLGA NPs in the nasal mucosa [100]. Similarly, a strong CTL response was induced by the inactivated swIV vaccine encapsulated in PLGA that was intranasally administered to pigs. The vaccinated pigs showed a significantly greater number of IFN-γ producing cells after virus re-stimulation when compared to mock vaccination. Furthermore, the vaccinated group demonstrated greater protection against the SwIV H1N1 challenge with less fever, lower viral titers, and reduced lung lesions [101]. Recently, Kole et al. (2019) studied the protective efficacy of PLGA encapsulated inactivated viral haemorrhagic septicaemia virus (VHSV) vaccine in olive flounder via the immersion route, followed by a booster through the oral route. The immunized group exhibited significantly higher levels of anti-VHSV antibodies in the fish sera, skin mucus as well as intestinal mucus in comparison to the naïve group. The study highlighted that PLGA encapsulated inactivated VHSV vaccine protected the antigens from enzymatic degradation through the oral route. In addition, it minimized stress in animals and therefore enhanced the overall vaccine efficacy [102].

Yang et al. (2021) investigated the cationic PLGA nanoparticles as an adjuvant for serotype-O of the foot and mouth disease (FMDV) DNA vaccine—pVAX-VP013 (Figure 5i). The plasmid vaccine was introduced separately with three different genes encoding for cytokine adjuvants: swine interleukin (IL)-2, IL-18, or granulocyte-macrophage colony-stimulating factor (GM-CSF) for comparison. Animal studies were carried out using guinea pig models. The results revealed that FMDV-specific neutralizing antibody levels were significantly higher in guinea pigs receiving PLGA-pVAX-VP013/IL-2, PLGA-pVAX-VP013/IL-18 and PLGA-pVAX-VP013/GM-CSF in comparison to guinea pigs receiving only the naked plasmid DNA. Lymphocyte proliferation assays also showed the FMDV vaccines co-administered with PLGA NPs were able to elicit enhanced cellular immunity in comparison with the traditional inactivated FMDV vaccine [103]. These studies indicated that the PLGA nanoparticles could be a promising biomaterial with adjuvant-like characteristics for the development of nanovaccines against viral infectious diseases.

Modifications of PLGA have been employed to enhance the immunogenicity of PLGA-NPs, such as coating with cationic polymer particles, including poly(ethyleneglycol) (PEG), chitosan or poly (ethylene imine) (PEI). As cells are negatively charged, these cationic particles could induce higher antigen uptake by the cells than the anionic PLGA particles alone. As such, the negatively charged *Angelica sinensis polysaccharide* (ASP) was encapsulated in PLGA coated with positively charged PEI and showed a significantly increased in the production of cytokines, including IFN-γ and L-12p70 [104]. These things considered, PLGA-NPs could be used to deliver multiple antigens or a combination of antigen-adjuvants to induce stronger mucosal and systemic immune responses. Toll-like receptors (TLRs) are known to be expressed in innate immune cells, including macrophages and dendritic cells, as well as non-immune cells, such as epithelial and fibroblast cells. The function of TLRs primarily recognizes microbial membrane proteins, proteins, and nucleic acids. TLRs have been studied extensively as vaccine adjuvants to enhance the proliferation of antigen presentation capabilities of innate immune cells. Encapsulation of these ligands in polymeric micro- or nanoparticles was demonstrated to provide a slow-controlled release system that could increase the antigen uptake as well as cross-presentations. For instance, incorporation of the TLR7/8 agonist (R848) or TLR9 agonist CpG oligodeoxynucleotide (CpG-ODN) to the PLGA nanoparticle-based vaccine encapsulating malarial antigen Pfs25 elicited higher innate cell activation as well as secretion of type I IFN-γ, thereby increasing the half-life of antibodies in rhesus macaques [105]. Similarly, Alkie et al. (2018) investigated the co-encapsulation of the inactivated avian influenza antigens (AIVs) with CpG-ODN 2007 adjuvant in PLGA against avian influenza. The results highlighted that PLGA encapsulating AIV and CpG-ODN2007 induced higher amounts of hemagglutinin-inhibiting antibodies in the vaccinated chickens when compared with the group vaccinated with non-adjuvanted AIV encapsulated in PLGA. Subsequently, chitosan-coated PLGA NPs that encapsulated inactivated avian influenza antigens (AIVs) coupled with the CpG-ODN 2007 adjuvant were developed (Figure 5iii). Induction of higher IgA and IgG antibody levels in sera and lachrymal secretions in the vaccinated chickens was observed compared to the group vaccinated with mannan-coated PLGA NPs encapsulating AIVs and the CpG-ODN adjuvant. The higher production of IgA and IgG induced by the chitosan-coated PLGA encapsulating AIVs and CpG-ODN was due to the mucoadhesive property of the chitosan as well as the prolonged interaction between the PLGA particles with the mucus which resulted in an increase in antigen uptake at the mucosal site [106].

## 8. Chitosan

Chitosan is a derivative of chitin found on the shells of crustaceans such as prawns and crabs. It is a naturally occurring polysaccharide, cationic and highly basic which is approved by the U.S. FDA for tissue engineering and wound healing functions [107,108]. Chitosan is well-known for its mucoadhesive properties that prevent nasal clearance as it interacts with the negatively charged mucus by forming a complex through ionic bonding or ionic interactions, therefore increasing the antigen retention time in the nasal mucosa [109]. This unique characteristic provided various administration routes for chitosan NPs including intravaginal, intranasal, intraocular, intrapulmonary, or intratracheal, which is essential to target the majority of viral diseases that enter the human body through mucosal pathways [110]. Chitosan can be an ideal nanomaterial used for DNA vaccine delivery due to its cationic nature, as it causes electrostatic binding to the anionic structure of DNA, leading to the formation of polymer-DNA complexes that would protect the DNA from enzyme degradation [111].

Additionally, chitosan nanoparticles display immunostimulatory capabilities by modulating dendritic cell maturation and induction of type 1 interferon (IFN) response through the STING (stimulator of interferon genes) pathway [112]. Chitosan NPs could activate the NLRP3 inflammasome due to its cationic nature. This led to caspase-1 activation, which in turn induced the production of cytokines, pro-interleukin-18 (pro-IL-18) and pro-IL-1β [48,113]. IL-1β is involved in facilitating immune cell translocation to the site of infected cells, while IL-18 is involved in mediating adaptive immunity via the production of interferon-gamma (IFN- γ) [114]. Caspase-1 is also known to induce the pyroptosis process, which suppresses intracellular viral replication leading to viral clearance [115].

Chitosan-based nanoparticles have been widely investigated in vaccine development against several infectious diseases, such as influenza and Newcastle disease [116,117]. Chitosan nanoparticles encapsulated inactivated whole influenza virus (Figure 6i) with CpG oligonucleotide (CpG ODN) or with the Quillaja saponin (QS) adjuvant have been shown to induce Th1 type responses in immunized rabbits. The dry powder nanosphere vaccine was administered intranasally to rabbits on days 0, 45, and 60, followed by intramuscular (IM) injection as the final booster on day 75. Both chitosan encapsulated whole inactivated virus with CpG (CH+ WV+ CpG) and chitosan encapsulated whole inactivated virus with QS (CH+WV+QS) stimulated significantly higher levels of IgG than the mock group, with the CH+WV+CpG group induced the highest IgG antibody level amongst all the groups. This finding is in line with the detection of the IgG levels where the CH+WV+CpG group induced the highest IgG antibody titer in the group immunized with this vaccine formulation. Similarly, the CH+WV+ CpG group elicited the highest sIgA titers in the nasal swab of the immunized rabbits. For the cellular mediated responses, a Th-1 biased response was observed in the group immunized with CH+WV+CpG as indicated by an increase in IL-2 and IFN-γ secretions. This could be attributed to the mucoadhesive effect of the chitosan and the adjuvant effect of the CpG. Thus, the vaccine formulation of chitosan encapsulated inactivated influenza whole virus with CpG serving as an adjuvant (CH+WV+CpG) elicited the best humoral and cellular mediated responses against influenza virus [118].

The enhanced protective efficacy of a Newcastle disease virus (NDV) DNA vaccine encapsulated in chitosan (pFNDV-CS-NPs) (Figure 6ii) was investigated in pathogen-free chickens [117]. Chickens administered with pFNDV-CS-NPs through intramuscular (IM) and intranasal (IN) routes demonstrated peak IgG levels at the fifth-week post immunizations and maintained relatively high IgG titers till the seventh week. In comparison, the IgG antibody titers of chickens immunized with the naked NDV DNA vaccine peaked at the fourth week post-immunization. Mucosal sIgA antibody titers were also found to be significantly higher while maintaining a longer sIgA secretion period in chickens administered with pFNDV-CS-NPs through the IN route when compared with other groups. Lymphocyte proliferation assays were carried out in chickens on 2-, 4- and 6-weeks post immunizations to assess the cell-mediated immune responses. pFNDV-CS-NPs were found to enhance significantly higher T lymphocyte immunity than the control group at the fourth- and sixth-week post IM and IN administrations. Therefore, it can be concluded that a stronger immune response was induced by pFNDV-CS-NPs.

Recently, Hajam and colleagues (2020) investigated the efficacy of chitosan encapsulating influenza mRNA, which was surface coated with conserved influenza antigens—HA2 proteins and M2e proteins. These NPs were able to elicit potent protective immunity in immunized chickens against influenza H7N9 and H9N2 viral challenges. The chickens were administered intranasally with thechitosan-coated-HAs-M2e vaccine, and it was observed that the vaccine candidate induced superior IgG and mucosal IgA antibody levels in immunized chickens. The immunized chickens were protected from viral infections by showing reduced viral shedding in feces and exhibited low lung pathology in comparison to the negative control group [116].

Additionally, chitosan could be chemically modified to bypass the restriction of its solubility at pH < 6.5 [119]. Several studies have shown poly-ethylene glycol (PEG), poloxamer, glycol, and alginate could be conjugated with chitosan for hydrophilic modifications, thereby enhancing water solubility and biocompatibility of such modified chitosans [120,121,122,123]. A hydrophobic derivative of chitosan, known as trimethyl chitosan (TMC), was also found to have excellent water solubility in a wide pH range in comparison with its parent compound. *N,N,N*-TMC NPs encapsulating Hepatitis B virus surface antigen (HBsAg) for controlled intranasal (i.n.) administration in mice was reported to induce significant IgG immune response in comparison with the two mice groups immunized with the HBsAg antigen co-administered with soluble *N*-TMC or the free antigen alone. Significant IgA antibody titers were only found in the nasal lavages of the mice group immunized with HBsAg-*N*-TMC NPs alone. N-TMC NPs were suggested to have boosted the residence and adhesion of HBsAg on the nasal mucosal membrane, thus making it a good adjuvant candidate [124].

Aside from chemical modifications to improve chitosan solubility and stability, modification by using ligands such as by conjugation of mannose, galactose, or peptides to chitosan carriers could improve cellular uptake and specific delivery of vaccines to target cells such as M cells [119]. M cells are found in the gut-associated lymphoid tissues (GALT), which help to regulate the infection and immunity of the GI tract [125]. A DNA vaccine (CPE30-CS-pVP1) designed with Coxsackievirus B3 (CVB3) predominant VP1 gene was orally delivered into mice using chitosan conjugated with M cell targeting ligand known as CPE30 peptide [126]. The IgG antibody titer and splenic T cell immune response induced by CPE30-CS-pVP1 remained the same in comparison with mice immunized with CS-pVP1. However, mucosal T-cell immunity and specific fecal SIgA levels against CVB3-induced myocarditis were significantly increased by CPE30-CS-pVP1 immunization. CPE30-CS-pVP1 immunized mice also demonstrated less severe myocarditis, and lower viral loads were observed.

## 9. Protein Nanoparticles (Ferritin)

Ferritin is naturally produced by many organisms, including bacteria, fungi, plants, and animals [127]. Ferritin plays an important role in iron homeostasis to ensure the bioavailability of iron in the human body [128]. The features of ferritin, such as self-assembly capacity, remarkable thermal and pH stability, biocompatibility, and biodegradability, render them ideal candidates for antigen display in vaccine development. In addition, the ferritin particles contain a hollow cavity in which can be engineered to encapsulate molecules such as peptides, drugs, and nucleic acids [129]. For instance, trimeric haemagglutinin (HA) encapsulated in ferritin nanoparticles (Figure 7i) resulted in the formation of particles with eight trimeric viral spike proteins on the surface of NPs. Immunization of mice with the nanovaccine elicited >10-fold higher hemagglutination inhibition (HAI) antibody titers than mice administered with the licensed inactivated influenza vaccine [130]. Other examples of multimerized self-assembly proteins, such as trimeric HIV-1 Env proteins displayed on the surfaces of ferritin nanoparticles, elicited higher levels of neutralizing antibodies against autologous tier 2 HIV virus in the immunized rabbits vaccinated with the ferritin nanoparticles compared to the rabbits immunized with the soluble trimers [131]. The study was carried out in both ferret and murine models. Yassine et al. (2015) engineered the HA stabilized stem (HA-SS) on the surface of ferritin nanoparticles (H1-SS-np) to investigate the immunogenicity in murine and ferret models. BALB/c mice were vaccinated three times with the H1-SS-np and empty nanoparticles on weeks 0, 8, and 11, followed by a lethal challenge against the highly pathogenic H5N1 influenza strain. At the end of the challenge, naïve mice and mice immunized with empty nanoparticles died from infections, whilst those immunized with the H1-SS-np vaccine survived. When the ferrets immunized with the empty nanoparticles were challenged at week 6 after the final boosting, they succumbed to infections, whereas the ferrets vaccinated with the trivalent inactivated vaccine and H1-SS-np vaccine survived. Despite the ability of the H1-SS-np vaccine to protect the ferrets against influenza mortality, it did not prevent infections. Interestingly, the authors suggested that there was a correlation between the HA stem antibody titers and the survival and body weight of the animals. To further investigate this correlation, immunoglobulins (Igs) from the H1-SS-np immunized mice were passively transferred to naïve mice 24 h pre-challenge with H5N1 influenza. Transferred Igs were able to neutralize group H1 subtypes (H1, H2, H5, and H9), and mice receiving immune Igs were completely protected from the lethal H5N1 challenge [132]. Currently, there are three ferritin-based HA stem vaccine candidates being investigated in the clinical phase I trial against influenza (clinical identifier NCT03814720, NCT03186781, NCT04579250). Results from the clinical trial with the clinical identifier (NCT03186781 indicated that the 60 µg dose in a single vaccine administration of the nanovaccine alone and the 60 µg dose administered in a prime-boost regime with an influenza DNA vaccine-elicited broadly neutralizing responses through the production of antibodies against the conserved HA stem of influenza. The nanovaccine candidate demonstrated good safety and was well tolerated by the vaccinated individuals [133].

In addition to influenza, ferritin has been utilized as one of the delivery platforms for the development of a nanovaccine against SARS-CoV-2. For example, Powell and others (2021) displayed the full-length ectodomain (S-Fer) of the spike protein or a C-terminal 70 amino acid deletion SΔC-Fer on the surface of ferritin NPs (Figure 7ii) and investigated the immunogenicity of the vaccine candidate in a murine model. A single dose immunization in mice elicited neutralizing antibodies in mice receiving the SΔC-Fer and induced higher levels of neutralizing antibody than mice that received only S-Fer. After the second dose of antigen, all groups of mice elicited detectable neutralizing antibodies, and mice immunized with SΔC-Fer had the highest overall neutralizing titers than the other groups. These results highlighted that the multivalent presentation of antigens on ferritin NPs facilitated higher elicitation of neutralizing antibodies against the Spike protein of SARS-CoV-2 [134]. Recently, Carmen and others (2021) evaluated the immunogenicity of SARS-CoV-2 spike-ferritin nanoparticle (SpFN) immunogen formulated with either Alhydrogel^®^ (AH) or Army Liposone formulation containing QS-21 (ALFQ) in a murine model. C57BL/6 mice were immunized by intramuscular injections with either SpFN+AH or SpFN+ALFQ as a single dose vaccine or in priming vaccinations followed by a boost at week 3. Mice immunized with SpFN+ALFQ were able to induce significantly higher Th-1 cytokine levels, including IFN-γ, IL-2, and TNF-α when compared to the group administered with SpFN+AH. Additionally, prime-boost vaccinations with SpFN+ALFQ resulted in more robust cytokine-producing T cells, which highlighted the two-dose regimen of SpFN +ALFQ vaccine formulation, which enhanced the generation of SARS-CoV-2 specific T-cell responses [135]. The vaccine candidate is being evaluated in a phase I clinical trial (Clinical identifier NCT04784767). Recently, a ferritin nanoparticle-based SARS-CoV-2 RBD vaccine was constructed by using the SpyTag/SpyCatcher technique. SpyTag is a peptide made up of 13 amino acid, and SpyCatcher is a protein consisting of a 116 amino acid complementary domain that was derived from the fibronectin-binding protein (FbaB) of *Streptococcus pyogenes* (Spy) [136]. The fusion of SpyTag/SpyCatcher occurred spontaneously upon mixing by forming isopeptide bonds, which could be used as an alternative strategy to covalent protein/antigen conjugation in nanovaccine application. A recent study demonstrated that the fusion of RBD-SpyTag to the Ferritin-SpyCatcher led to the formation of ferritin-NP-RBD. in order to evaluate the immunogenicity of the nanovaccine, C57BL/6 mice were administered with the nanovaccine or equimolar RBD-SpyTag in combination with CpG-1826 adjuvant as a control group. Mice immunized with the ferritin-NP-RBD elicited potent antibody responses, which were approximately 100-fold higher than the control group. Additionally, the antibody levels of the mice immunized with the ferritin-NP-RBD were maintained significantly higher than the RBD-SpyTag group for at least 7 months, suggesting that the nanovaccine induced durable antibody immunity [137].

## 10. Non-Biodegradable Nanoparticles

### Calcium Phosphate (CaP) NPs

Calcium phosphate (CaP) is biocompatible, bioresorbable, and safe to be used as a vaccine adjuvant and in delivery vehicles. CaP was approved by the FDA for in vivo use in 2002 for lower spine surgery, and it was found to be safe without inducing cytotoxic effects when it was administered subcutaneously in a phase I study [138,139]. CaP NPs are well tolerated in the human immune system due to their chemical similarity with calcium phosphate, which is naturally occurring in the human body [140]. The most common CaP NP used as a carrier in medical applications is known as hydroxyapatite, which is the least soluble and most stable phase of all calcium phosphate compounds [141]. CaP is a poor microbial growth substrate which translates to its excellent storage stability. CaP is also relatively stable in human blood (pH 7.4) and is less soluble in a more alkaline environment [142]. However, the solubility of CaP increases in a low pH environment which allows targeted biodegradation and release of active molecules upon reaching inflammatory or microbial residing sites, as well as the endo-lysosome which is the entry pathway for some viruses including coronavirus [143,144]. There have been studies suggesting CaP NPs to be a valid alternative to replace the alum adjuvant due to its ability to induce higher IgG titers and triggering less significant IgE responses [140,145,146].

A previous study reported a comparison of CaP nanoparticles and microparticles adsorbed with the inactivated Human Enterovirus 71 (HEV-71) (Figure 8) with the inactivated virus (IV) vaccine alone in rabbit models through intramuscular (IM) and intradermal (ID) routes of administrations [147]. HEV-71 specific IgM and IgG antibody responses post immunizations (weeks 1, 3, 5, 7, and 9) were measured using ELISA. IgM antibody levels of rabbits immunized with either nano-CaP or micro-CaP adsorbed vaccines were significantly elevated through both IM and ID administration routes after the third vaccination dose when compared with the IV alone. Rabbits immunized with Cap-NPs adsorbed vaccine also displayed higher levels of IgG antibody titers in comparison with micro-sized Cap adsorbed vaccine and the IV alone. Cell viability was used to compare the HEV-71 neutralizing antibody titer of rabbit sera between different immunization groups. Rabbits immunized with the nano-CaP adsorbed vaccine administered through IM and ID routes showed the highest antibody titer (1:1600), followed by vaccine adsorbed to micro-CaP (1:800). In comparison, rabbits administered with the inactivated vaccine alone through IM and ID routes displayed low neutralizing titers of 1:400 and 1:200, respectively.

A total of six DENV peptides specifying multi-epitopes derived from the capsid and non-structural proteins (NS48, NS5, and NS2A) were administered to HLA-A2+ transgenic mice by using calcium phosphate nanoparticles (CaPNP) as a carrier. Mice immunized with CaPNP multi-peptides elicited higher CD8^+^ T cell responses when compared with the group of mice immunized with multi-peptides mixed with the standard adjuvant-ISA51, indicating that CaP NPs alone could act as an antigen delivery system as well as an adjuvant which facilitated antigen uptake by APC, thereby stimulating the activation of T cells [148].

The CaP NP surface consists of positively charged Ca^2+^ and negatively charged PO_4_^3−^ sites which could serve as good binding or conjugating sites for polymers such as poly (ethylene glycol) (PEG) and polyethyleneimine (PEI) [149,150,151]. Surface modifications of CaP NPs with hydrophilic or hydrophobic molecules would be able to alter the charge and increase the stability of CaNPs in vaccine applications [142]. Chemical moieties including COOH^−^, NH_2_^−^ or OH^−^ that were provided by conjugation to these polymers would also allow adsorption or adhesion of additional molecules such as DNA, siRNA, TLR ligands or peptides on the surface of CaNPs, thereby increasing their immunogenic efficacy [147].

Since CaPNPs were shown to elicit both humoral and cellular-mediated immunity, they provide great potential as a new platform in vaccine development. However, limitations such as rapid aggregation of the nanoparticles have to be overcome during the manufacturing process [140].

**Table 1 pharmaceutics-14-02554-t001:** Nanoparticle-based vaccines against infectious diseases in clinical trials.

Infectious Agents	Vaccine Candidates	Nanoparticles	Clinical Identifier	No. of Participants	Clinical Trials	References
COVID-19	ARCT-021	Lipid NPs	NCT04480957	106	Phase I//II (completed)	[152]
	BNT162b2	Lipid NPs	NCT04760132	10,000	Phase IV (Recruiting)	[152]
	Covac 1	Lipid NPs	ISRCTN17072692	192	Phase I (completed)	[153]
	ChulaCov19	Lipid NPs	NCT04566276	222	Phase I/II (Active, not recruiting)	-
	mRNA-1273	Lipid NPs	NCT04760132	10,000	Phase IV (Recruiting)	-
	mRNA-1273.351	Lipid NPs	NCT04785144	135	Phase I (Active, not recruiting)	-
	SpFN	Ferritin-NPs	NCT04784767	29	Phase I (Active, not recruiting)	[154,155]
Influenza	HA ferritin	Ferritin-NPs	NCT03186781	50	Phase I (completed)	[133]
HIV	MPER-656	Liposome	NCT03934541	24	Phase I (completed)	[156]

Note: SpFn: Spike-Ferritin-Nanoparticle.

**Table 2 pharmaceutics-14-02554-t002:** Advantages and disadvantages of various nanoparticles.

Nanoparticle	Advantages	Disadvantages	References
Liposome	Biodegradable, biocompatible, lower toxicity Approved by FDA as one of the earliest nano-drug Mimics cell membrane structure, less safety concerns Provide good entrapment for both hydrophilic and hydrophobic antigens Facilitate controlled release and protection of payload from GI environment	High production cost Low solubility in water Short half-life in human body Relatively large size: unstable in the GI environment poor permeability across epithelial of GI Low loading capacity for hydrophobic agents due to the limited space of lipid domain	[92,93,157,158,159,160,161,162,163]
PLGA	Biodegradable, biocompatible Approved by American FDA Can encapsulate a large variation of molecules Acting adjuvant, stimulates DC maturation Easily modified for functional and specific cell targeting improvements	Anionic nature: (1) low encapsulation efficiency of anionic Ag such as pDNA; (2) require modification of surface charge for rapid uptake by negatively charged cell membrane Initial burst release profile Relatively short systemic circulation time, inability to pass through the blood brain barrier (BBB)	[97,164,165,166,167,168,169,170,171]
Chitosan	Biodegradable, biocompatible Mucoadhesive properties allowing mucosal delivery routes and longer antigen retention time Facilitate penetration of epithelium through tight junctions	Bad solubility in water at neutral or basic pH Easily degraded in low pH media such as gastric acid positive charge could destabilize the cell membrane, causing cytotoxicity	[109,110,172,173,174,175,176]
Ferritin	Biocompatible Allows uniform display of 24 separate epitopes/peptides on the surface	Nanoparticle heterogeneity Insufficient antigen folding or interactions between subunits leading to antigen interference	[177]
Calcium phosphate	Biodegradable, biocompatible Non-toxic due to Cap naturally occurring in human body Excellent storage Relatively stable in human blood at neutral pH High affinity to nucleic acid for entrapment	Upscaling difficulties Particle agglomeration Synthesis method may require high end laser equipment	[140,142,178,179,180,181,182,183,184,185]

## 11. Future Prospectives and Outlook

For the past two decades, nanotechnology has facilitated the development of new vaccines against highly infectious diseases such as HIV, influenza, and COVID-19. Compared with traditional vaccines, nanovaccines utilize a variety of nanoparticles with advantages such as delivery efficiency, adjuvancity, dose regimes, and administration routes. Other benefits of nanovaccines include their antigenicity, controlled release, and biodegradable characteristics, which allow them to elicit both the humoral and cell-mediated immune responses along with memory effector responses, thereby alleviating the need for frequent boosters as in the traditional vaccine regimes. Although there are many nanovaccines being evaluated in the pre-clinical studies that could potentially become the new generation nanovaccines for the preparedness of future pandemics, the slow adoption of nanoparticles as new-generation adjuvants could be attributed to safety concerns, manufacturing, and scaling-up processes as well as regulatory frameworks for nanovaccine productions. Safety is the biggest public concern when it comes to the development of new vaccines and vaccine adjuvants, as there were few studies that indicated that inorganic nanoparticles might possess inherent toxicity after prolonged exposures, and cytotoxicity of nanoparticles was shown to be closely related to nanoparticle dosages. Furthermore, since nanoparticles have a relatively short history of applications in medicine and they do not have a longstanding safety profile in human vaccinations, it is essential to study the safety of nanoparticles for the delivery of novel vaccines in animal models extensively before proceeding to human testing to demonstrate the safety of materials and adjuvant properties before proceeding to clinical trials. Biodegradable components such as liposomes and lipid-nanoparticles have played dominant roles in nanovaccine clinical applications due to their excellent biocompatible profiles. This suggested that the biodegradable and biocompatible profiles of nanoparticles are vital factors to consider in the development of next generation nanovaccines. Another barrier to nanovaccine development is the manufacturing and scaling-up processes. After the manufacturing process, inconsistency of the physiochemical properties of the nanoparticles, such as size distributions, surface charges, encapsulation/ conjugation efficiencies, and antigen release profiles, could cause unexpected non-specific immune responses after biodegradation. Other than that, in the pre-clinical studies, small-scale laboratory research should provide the capability to easily scale up the production of nanovaccines, but this could become an issue when there is the need to scale up in a sterile environment. A lack of guidelines for the regulatory requirements of nanovaccine production has caused a high degree of uncertainty for vaccine developers. Hence, a stringent regulatory framework is required to ensure the quality, safety, stability profiles, and efficacy of nanovaccines.

Although much work is necessary to facilitate the clinical translation and applications of nanovaccines for human vaccinations, we are now better equipped with the experience gained from the COVID-19 pandemic and the successful clinical application of the lipid-nanoparticle-based Pfizer/BioNTech and Moderna mRNA COVID-19 vaccines which highlighted the promising future of nanovaccines against various infectious diseases.

## Figures and Tables

**Figure 1 pharmaceutics-14-02554-f001:**
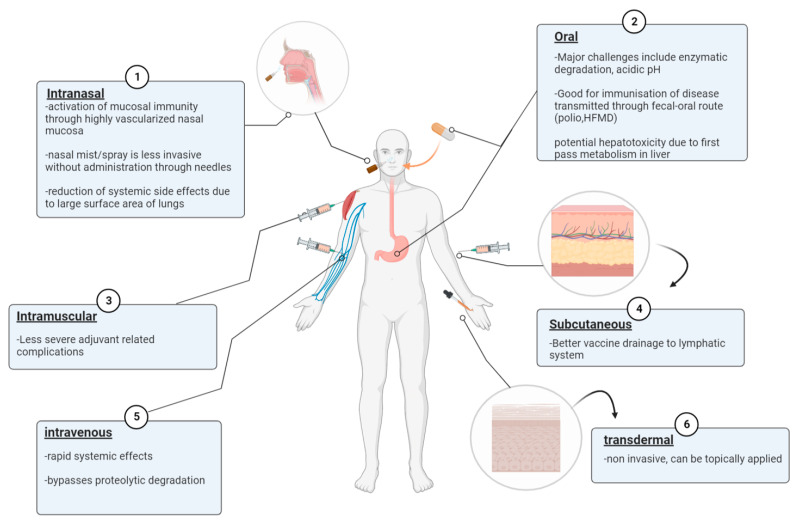
Different vaccine administration routes against infectious diseases including intramuscular (IM), subcutaneous (SC), oral, intranasal, and intravenous (IV).

**Figure 2 pharmaceutics-14-02554-f002:**
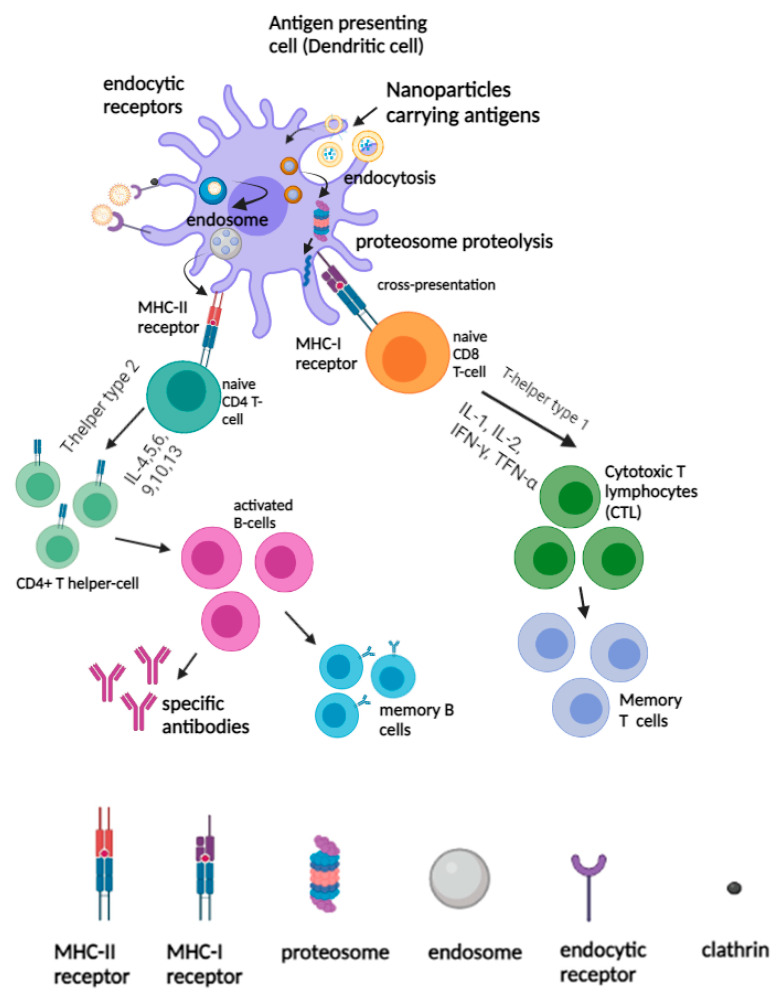
Activation of innate and adaptive immunity by nanovaccines. The uptake and presentation of nanovaccine by APCs (for example, dendritic cells) elicited humoral and cellular immune responses.

**Figure 3 pharmaceutics-14-02554-f003:**
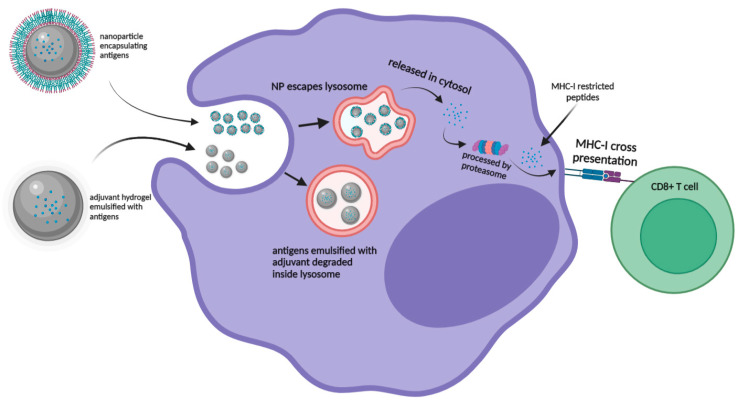
Graphic representation of nanoparticle encapsulated antigens escaping lysosome and processed through “cytosolic” pathway for MHC-I presentation for CD8^+^ T cells in comparison with adjuvant emulsified antigens degrading within the lysosome.

**Figure 4 pharmaceutics-14-02554-f004:**
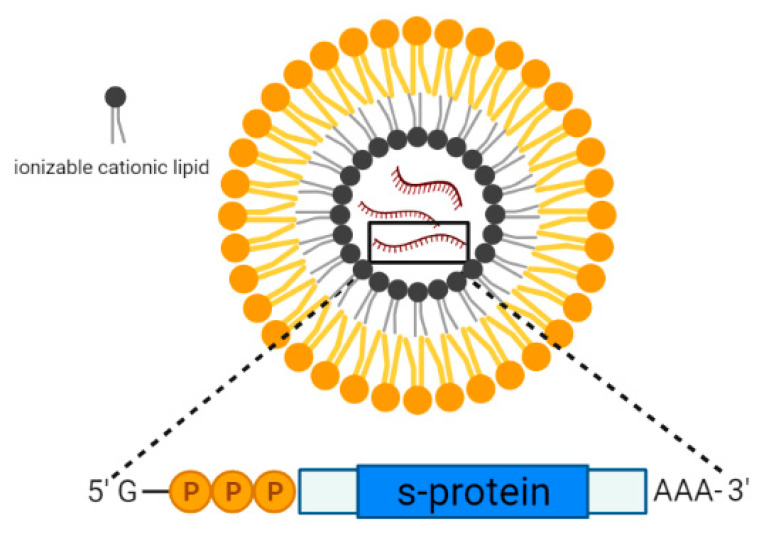
Schematic representation of the lipid nanoparticle consisting of ionizable cationic lipids encapsulating mRNA encoding the SARS-CoV-2 spike protein.

**Figure 5 pharmaceutics-14-02554-f005:**
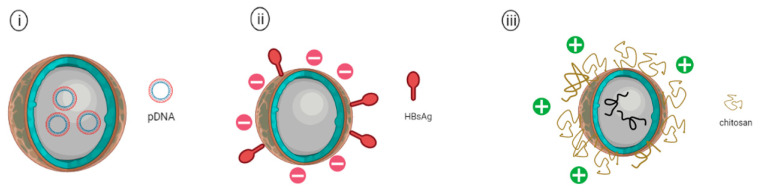
Schematic representation of (**i**) PLGA nanoparticles encapsulating plasmid DNA vaccine, (**ii**) PLGA nanoparticle adsorbed with hepatitis B surface antigen (HBsAg) on the surface of PLGA NP, and (**iii**) Chitosan coated PLGA nanoparticle encapsulating avian influenza antigens.

**Figure 6 pharmaceutics-14-02554-f006:**
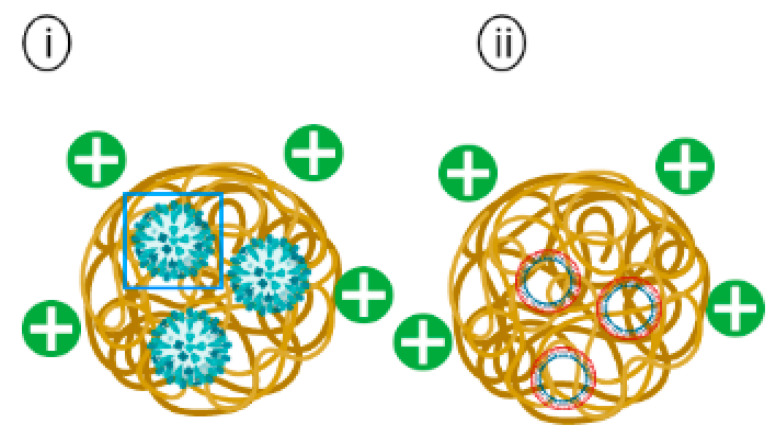
Schematic representation of (**i**) Chitosan encapsulating inactivated whole influenza virus and (**ii**) Chitosan encapsulating plasmid DNA vaccine.

**Figure 7 pharmaceutics-14-02554-f007:**
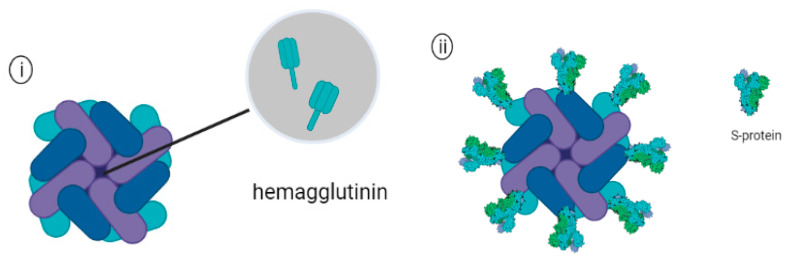
Schematic representation of (**i**) Ferritin encapsulating trimeric hemagglutinin (HA) and (**ii**) Ferritin attached with Sars-Cov-2 spike proteins.

**Figure 8 pharmaceutics-14-02554-f008:**
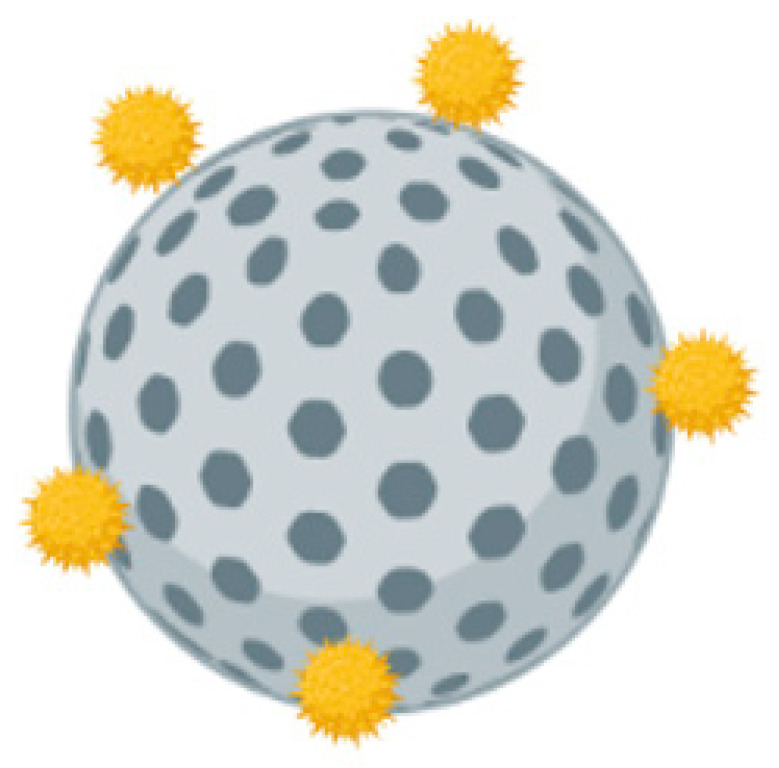
Schematic representation of calcium phosphate nanoparticle adsorbed with antigens.

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
