# Peer review of "Nanovaccines against Viral Infectious Diseases"

_pharmaceutics, 2022, doi:10.3390/pharmaceutics14122554_

Round 1

Reviewer 1 Report

The authors made an interesting literature revision regarding vaccine approaches for infectious diseases. The manuscript make more emphasis on the nanocarriers than on the targeted diseases and the vaccination strategies. Nevertheless, the work is well prepared. 

I have only one comment to be emphatic about. Making a revision about novel vaccination approaches is valuable. However, there is not real mention of the novel mRNA technologies and the impact this technology is generating in the field of vaccinology. I would recommend including a section about this novel and promising technology. 

Author Response

Thank you.

Reviewer 2 Report

The present review entitled "Nanovaccines against viral infectious diseases" describes various nanoparticle-based vaccine formulations against viral infections. The manuscript  is well written and briefly describes various vaccine delivery systems. However, the authors should revise the manuscript in order to make it acceptable for its publication in pharmaceutics.

1. The authors should include a figure showing a difference in the processing antigens delivered through nanoparticles or emulsified in other adjuvant, particularly in the context of CTL-mediated immune response that is critical against viral infectious diseases.

2. The authors should include one table showing advantages and disadvantages of various nanoparticle-based vaccine delivery systems.

3. The incorporation of immune-stimulatory molecules is an important strategy to enhance the efficacy of antiviral vaccine. This should be included in the current manuscript. The following recent references should be included:

1. Role of NKT Cells during Viral Infection and the Development of NKT Cell-Based Nanovaccines. Vaccines (Basel). 2021 Aug 26;9(9):949. 

2. Encapsulation of MERS antigen into α-GalCer-bearing-liposomes elicits stronger effector and memory immune responses in immunocompetent and leukopenic mice. J King Saud Univ Sci. 2022 Jul;34(5):102124.

Author Response

Additional information regarding the immunostimulatory nature of chitosan nanoparticles have been included as supplementary information as well.

Thank you

Reviewer 3 Report

Manuscript ID: pharmaceutics-1994620

The manuscript describes a well performed review on nanovaccines against viral infectious diseases. Various nanoparticle based-vaccines have been clearly documented through their composition, vaccination strategies and effectiveness against viral challenge in several preclinical models as well as in clinical trials. Especially the current COVID-19 pandemic and the speed of generating effective RNA vaccines against SARS-CoV-2 has clearly set the spot lines on the development of these next generation vaccine candidates. The review is acceptable for publication.

Some minor comments:

·      I miss Table1. Otherwise change Table 2 (ln 50 and 65 and ln 454) into Table 1.

·      Remove the P values in ln 295 and 297 since significancy is already mentioned in the same sentences.

·      Ln 339-341 describes and compared the immunization of both ferrets and mice in the same study??

·      Remove the abbreviation “(HA)” in ln 406 as this might be confusing because throughout the manuscript the authors have used HA already for the Influenza HA (hemagglutinin) antigen.

·      Remove “(MPER)” in the footnote of the table.

Author Response

Thank you

Round 2

Reviewer 2 Report

The authors have responded to all comments.